# Hypoxia-Inducible Factor-1α, a Novel Molecular Target for a 2-Aminopyrrole Derivative: Biological and Molecular Modeling Study

**DOI:** 10.3390/cancers18010115

**Published:** 2025-12-30

**Authors:** Svetlana S. Zykova, Tatyana Gessel, Aigul Galembikova, Evgenii S. Mozhaitsev, Sophia S. Borisevich, Nazim Igidov, Emiliya S. Egorova, Ekaterina Mikheeva, Natalia Khromova, Pavel Kopnin, Alina Galyautdinova, Vladimir Luzhanin, Maxim Shustov, Sergei Boichuk

**Affiliations:** 1Perm State Pharmaceutical Academy, Perm 614990, Russia; zykova.sv@rambler.ru (S.S.Z.); igidov_nazim@mail.ru (N.I.); vladimir.luzhanin@pharminnotech.com (V.L.); mshustov1@mail.ru (M.S.); 2Department of Pathology, Kazan State Medical University, Kazan 420012, Russia; ivoilova.tatyana@mail.ru (T.G.); ailuk000@mail.ru (A.G.); miheeva.1973@bk.ru (E.M.); galyautdinovaalina2003@yandex.ru (A.G.); 3N.N. Vorozhtsov Novosibirsk Institute of Organic Chemistry, Siberian Branch of Russian Academy of Sciences, Novosibirsk 630090, Russia; mozh@nioch.nsc.ru; 4Ufa Institute of Chemistry, Ural Federal Research Center, Russian Academy of Sciences, Ufa 450054, Russia; monrel@mail.ru; 5Central Research Laboratory, Kazan State Medical University, Kazan 420012, Russia; jastspring@yandex.ru; 6Cytogenetics Laboratory, Carcinogenesis Institute, N.N. Blokhin National Medical Research Center of Oncology, Moscow 115478, Russia; n.khromova@ronc.ru (N.K.); pbkopnin@mail.ru (P.K.); 7Department of Radiotherapy and Radiology, Faculty of Surgery, Russian Medical Academy of Continuous Professional Education, Moscow 125993, Russia

**Keywords:** HIF-1α, 2-aminopyrroles, heterocycles, apoptosis, proliferation, tumor growth, folding, molecular docking

## Abstract

Given that hypoxia-inducible factor-1α (HIF-1α) plays a significant role in cancer development and progression, it has been an intriguing therapeutic target for cancer research. We show here that a 2-aminopyrrole derivative (2-amino-1-benzamido-5-(2-(naphthalene-2-yl)-2-oxoethylidene)-4-oxo-4,5-dihydro-1-H-pyrrole-3-carboxamide—2-ANPC), a microtubule-binding agent, also targets HIF-1α and effectively downregulates HIF-1α on both transcriptional and translational levels. This effect was observed in a broad spectrum of epithelial cancer cell lines in vitro and in vivo by using the breast cancer syngraft model. The latter was also associated with decreased expression of receptors for vascular endothelial growth factors and enhanced intratumor apoptosis, which, in turn, led to reduced tumor weight and volume. Overall, 2-ANPC can serve as a scaffold for the development of successful chemotherapeutic anticancer agents with dual therapeutic modalities.

## 1. Introduction

Hypoxia-inducible factor HIF-1α is a well-known regulator of cellular responses to hypoxia, functioning by activating several genes. Under normoxic conditions, HIF-1α becomes hydroxylated in the oxygen-dependent degradation (ODD) domain and, after recognition by the von Hippel–Lindau tumor suppressor protein (pVHL), is degraded by the ubiquitin–proteasome pathway. Under hypoxic conditions, HIF-1α expression is usually upregulated. This, in turn, is often associated with its translocation into the nucleus, where it accumulates, dimerizes with HIF-1β (aryl hydrocarbon receptor nuclear translocator—ARNT), and, after binding to CREB-binding protein (CBP) or p300 activators, interacts with hypoxia-responsive elements to induce gene expression.

The expression of HIF initiates the mechanism of tissue adaptation to hypoxia. This leads to enhanced erythropoiesis, increasing the oxygen-carrying capacity of blood; triggers angiogenesis via activation of vascular endothelial growth factor (VEGF), thereby expanding the blood supply to hypoxic regions; and regulates glucose metabolism, reducing oxygen demand for adenosine triphosphate production. The role of the HIF is also crucial in signaling pathways associated with both cell proliferation and the maintenance of normal tissue apoptosis [1].

The consequences of HIF expression are highly significant, not only in hypoxic normal tissues and organs but also in tumor development. Significant cell density and increased metabolic activity, coupled with insufficient oxygen supply [2], lead to HIF involvement in tumor growth and development, thereby enhancing blood supply.

HIF-driven metabolic changes sustain viability under pathological conditions in which reduced oxygen levels reduce the efficacy of both radiotherapy and chemotherapy as available oxygen reserves are insufficient to counter oxidative damage to cells [3]. Furthermore, mechanisms that confer cell survival under hypoxic conditions also contribute to drug resistance, for example, by facilitating the active efflux of chemotherapeutic agents (e.g., doxorubicin) from cancer cells [3].

The HIF-mediated pathway is deeply involved in tumor cell evasion of apoptosis. On one hand, p53 protein, a well-known tumor suppressor, inhibits HIF activity and induces apoptosis upon activation [4]. Conversely, p53 is downregulated by the overexpression of HIF-1α due to the hypoxic tumor microenvironment (TME). Indeed, increased expression of HIF-1α reduces p53 levels and attenuates its transcriptional activities by competing for p300, a coactivator of both p53 and HIF-1α [5]. Additionally, p53 mutations leading to loss of function are known to be a common genetic alteration in human cancers observed in more than 50% of cancer cases [6]. This results, in particular, in the absence of p53-mediated apoptosis and an increased role for HIF in oncogenesis. Moreover, recent studies suggest that p53 mutations can induce the sustained activation of the mTOR pathway, which is a positive regulator of HIF-1α [7]. Besides its potency in inhibiting function of p53, HIF reduces the activity of the apoptosis regulator Bcl-2 [8].

Thus, HIF-1α is considered an attractive target for cancer therapy, thereby making the development of strategies to inhibit HIF-1α activity and its associated pathways a high priority. In particular, this includes the approaches to inhibiting HIF-1α transcriptional activity, translation, stability, heterodimerization, nuclear transport, DNA binding, and HIF target genes [4,9]. For example, anthracyclines, topoisomerase inhibitors, steroids, and microtubule-targeting agents (MTAs) are shown to inhibit HIF-1α mRNA expression [10]. Several drugs with distinct therapeutic activities can also effectively block HIF-1α mRNA translation. This includes mTOR inhibitors (e.g., rapamycin and everolimus), MTAs (e.g., taxotere), cyclooxygenase type 2 inhibitors (ibuprofen), topoisomerase inhibitors (e.g., mitoxantrone and topotecan), cardiac glycosides (e.g., digoxin), and the inhibitors of heat shock protein 90 (HSP90) (e.g., ganetespib) [11,12]. Similarly, some HSP90 inhibitors (e.g., 17-allylamino-17-demethoxygeldanamycin) were shown to reduce HIF-1α protein stability and promote its proteasomal degradation. This activity was also observed for class II histone deacetylase (HDAC) inhibitors (e.g., panobinostat) and the thioredoxin inhibitor (e.g., PX-12) [13]. Several reports indicate that HDAC inhibitors can stabilize the HIF-1α protein and promote HIF-1α nuclear localization via modulating the acetylation of either the HIF-1 protein itself or of its cofactors (e.g., p300) [14,15,16]. Anthracyclines (e.g., doxorubicin and daunorubicin) are widely used in cancer treatment due to their ability to inhibit topoisomerase activity. Besides this fact, they were found to decrease HIF-1 transcriptional activity via the inhibition of HIF-1 heterodimer binding to the hypoxia response element DNA sequences [17].

We examined here the potential of 2-ANPC to interact with and inhibit HIF-1α activity in vitro and in vivo across a broad spectrum of cancer cell lines. We show here, for the first time, that 2-ANPC interacts with HIF-1α and promotes its proteasome-mediated degradation, resulting in decreased HIF-1α expression in cancer cells both in vitro and in vivo. Moreover, reduced HIF-1α expression in syngraft tumors moderately downregulated VEGFR1 and 3 and was associated with decreased tumor volume and size. Using various computational tools, we confirmed the effective targeting of HIF-1α by 2-ANPC and identified potential binding sites for 2-ANPC to interact with HIF-1α, HIF-1β, and the p300 complex.

Collectively, we show here that the 2-ANPC synthesized in our lab and previously shown to be an effective MTA [18,19] also exhibits potent anti-HIF-1α activity, thereby demonstrating this derivative as a scaffold for the development of successful chemotherapeutic anticancer agents with dual therapeutic modalities.

## 2. Materials and Methods

### 2.1. Chemistry

The synthesis of 2-ANPC was described earlier in detail (Figure 1) [20]. Paclitaxel (PTX) and MG-132, a proteasome inhibitor, were purchased from Sigma-Aldrich (St-Louis, MO, USA), whereas cobalt chloride (CoCl_2_) was obtained from Wako (Osaka, Japan).

### 2.2. Cell Lines and Culture Conditions

Triple negative breast cancer HCC1806, MDA-MB-231, and 4T1; non-small cell lung carcinoma H1299; and prostate cancer PC-3 cell lines were obtained from the American Type Culture Collection (ATCC, Manassas, VA, USA). These cancer cells were cultured in RPMI-1640 medium (PanEco, Moscow, Russia) mixed with 15% fetal bovine serum (Biosera, Cholet, France), 50 U/mL penicillin, and 50 μg/mL streptomycin (PanEco, Moscow, Russia). Cell lines were maintained in a CO_2_ incubator (LamSystems, Mass, Russia) with high humidity and CO_2_ level at 37 °C.

### 2.3. Real-Time Monitoring of Cell Proliferation

E-Plate L8 PET cassettes (ACEA Biosciences, San Diego, CA, USA) were used to study proliferation in a real-time cell analyzer iCELLigence (ACEA Biosciences, San Diego, CA, USA). Cancer cells were exposed to DMSO (negative control), 0.5 mM PTX (positive control), or 5 mM or 10 mM 2-ANPC on the day after seeding into cassettes. The proliferation rate was recorded every hour, and a graph of growth kinetics was formed in real time. Three independent experiments were performed. The normalized cell index was used for results analysis in RTCA Software version 1.0 (ACEA Biosciences, Inc., San Diego, CA, USA).

### 2.4. Antibodies

Primary antibodies raised against the following proteins were used for Western blotting and IHC-staining: cleaved forms of caspase-3 (#9662S) and PARP (#5625S) (Cell Signaling, Danvers, MA, USA); actin (A00730-200, Abcam, Cam-bridge, MA, USA); and HIF-1α (sc-10790), VEGF-1 (sc-271789), and VEGF-3 (sc-28297) (Santa Cruz Biotechnology, Santa Cruz, CA, USA). HRP-conjugated secondary antibodies for Western blotting were purchased from Santa Cruz Biotechnology.

### 2.5. Western Blotting

Cells were washed out twice in 1X phosphate-buffered saline (PBS) and incubated in RIPA buffer (25 mM Tris-HCl, pH 7.6, 150 mM NaCl, 5 mM EDTA, 1% NP-40, 1% sodium deoxycholate, 0.1% SDS) with protease inhibitor cocktail solution and PMSF (Sigma-Aldrich, St-Louis, MO, USA) for 20 min at 4 °C. Whole-cell lysates were obtained by centrifugation for 30 min at 13,000 rpm at 4 °C. Lysates normalized using a BCA assay (Thermo Fisher Scientific, Rockford, IL, USA) to a protein concentration of 30 μg were added to 4–12% Bis-Tris NuPAGE gel (Invitrogen, Carlsbad, CA, USA). Gels were run at low constant voltage (80 V) in 1X NuPAGE MOPS SDS running buffer (Invitrogen, Carlsbad, CA, USA). After separation, proteins were transferred to nitrocellulose membrane using 1× transfer buffer (25 mM Tris, 192 mM Glycine, 20% (*v*/*v*) methanol, pH 8.3) at 350 mA for 1.5 h at 4 °C. Membranes were incubated in antibody-dilution buffer to reduce non-specific binding (1 h, room temperature), primary (overnight, 4 °C), and secondary (1 h, room temperature) antibodies consistently according to the manufacturer’s protocol. Detection of results was performed on the Fusion Solo S imaging system (Vilber Lourmat, Collégien, France). Three independent experiments were performed. Protein expression was measured by NIH Image J software (Version 1.49, Bethesda, Rockville, MD, USA) for densitometry analysis.

### 2.6. RNA Extraction and RT-PCR

Complementary DNA (cDNA) was reverse-transcribed from RNA extracted from cancer cells according to the protocol described [21]. RT-PCR was performed with 1 µL of template cDNA, 5× qPCRmix-HS SYBR (PB025, Evrogen, Moscow, Russia), and 10 mM of forward and reverse primers for HIF-1α or GAPDH genes (Appendix A) using the CFX96 Real-Time Detection System (Bio-Rad, Hercules, CA, USA). GAPDH was used to normalize HIF-1α levels and obtain relative levels. The number of cycles required to increase the fluorescence signal to a specific detection threshold (Ct value) showed the quantitative data.

### 2.7. In Vivo Study of Antitumor Activity

The animal experimental protocols were approved by the Committee for Ethics of Animal Experimentation, and the experiments were conducted in accordance with the Guidelines for Animal Experiments in N.N. Blokhin National Medical Research Center of Oncology. The syngeneic 4T1 breast cancer mouse model was used to analyze the antitumor activity of 2-ANPC. After subcutaneous transplantation of cancer cells into the flank regions of Balb/c mice and tumor growth to a volume of 200 mm^3^, 10 animals were divided into two groups: control (*n* = 5) and 2–ANPC–treated (*n* = 5). Administration of vehicle (control) or 10 mg/kg of 2-ANPC was carried out on 10, 13, 16, 19, 22, 25, and 28 days. Animals were sacrificed on the 30th day. Isolated tumors were weighed to obtain the terminal weight and measured using a caliper to calculate terminal volume (width × height × depth × 0.5). Formalin-fixed, paraffin-embedded (FFPE) tissues were sectioned at 4 μm. Prepared samples were stained with H&E. Immunohistochemical staining was used to detect the expression of HIF-1α, VEGFR-1, VEGFR-3, and the cleaved form of caspase-3. Representative images were obtained using an Olympus BX63 microscope (Olympus, Tokyo, Japan). A total of 100 cells were counted in every two sections from 10 random fields of view to determine the number of cancer cells (%) with the desired phenotype. Data are presented as mean ± standard deviation for each group. Differences were considered significant at *p* < 0.05 (*) using an unpaired Student’s *t*-test.

### 2.8. Statistics

All the experiments were repeated a minimum of 3 times. The Shapiro–Wilk test was used to assess normality. Normally distributed data are presented as mean ± standard deviation for each group. Differences were considered significant at *p* < 0.05 (*) using an unpaired Student’s *t*-test.

### 2.9. HIF-1α Modeling

The protein sequence of HIF-1α was taken from UniProtKB Q16665-1, and its structure was predicted with AlphaFold 2 Colab [22,23] for the following molecular dynamics system, which was built in Visual Molecular Dynamics (VMD) v1.9.3 [24]: TIP3P water and 0.015 M NaCl. The simulation was performed for 100 ns in Nanoscale Molecular Dynamics (NAMD) v2.15 [25,26] using the CHARMM36 force field [27,28,29]. After the HIF-1α dynamics simulation, the 99.6 ns snapshot was chosen for the docking based on the Ramachandran plot data.

For the multiligand simulation, 26 copies of I were added to the solvate box, and the simulation was performed for 300 ns. The system was built in VMD v1.9.3 [24] (TIP3P water, 0.015 M NaCl), and the simulation was performed using NAMD v2.15 [25,26] using the CHARMM36 force field [27,28,29]. The ligand parametrization was performed with SwissParam [30,31].

### 2.10. HIF-1 Protein Complex Modeling

The protein sequences of HIF-1α, HIF-1β, and p300 were taken from UniProtKB (Q16665-1, P27540-1, and Q09472, respectively). The HIF-1 complex was predicted with AlphaFold 3 [32]. The following molecular dynamics simulation was performed for 300 ns in Desmond [33] by using the OPLS4 force field [34]: TIP3P water and 0.015 M NaCl. The trajectory was clustered, and the snapshot of the system for the docking was chosen based on the Ramachandran plot data.

### 2.11. Site Mapping and Docking

All the molecular dynamics snapshots were minimized by the OPLS4 force field [35] in Prime [36,37] before docking. The search for binding sites was performed by SiteMap [38,39]. Molecular docking was performed using the forced ligand positioning protocol (energy spent on formation of the laying of the compound in the binding site and binding energy of ligand and protein—IFD) [40,41] with the following conditions: flexible protein and ligand; grid matrix size of 20 Å; amino acids (within a radius of 5 Å from the ligand) restrained and optimized, taking into account the influence of the ligand; and the maximum number of positions limited to 10. Docking solutions were ranked by evaluating the following calculated parameters: docking score (based on Glide score minus penalties) and the parameter of model energy value (Emodel), including the Glide score value, energy unrelated interactions, and the parameters of energy spent on formation of the laying of the compound in the binding site and binding energy of the ligand and protein (IFD score).

## 3. Results

### 3.1. 2-ANPC Exhibits Potent Anti-Proliferative and Pro-Apoptotic Activities Against Breast, Lung, and Prostate Cancer Cell Lines

We initially examined whether 2-ANPC affects the proliferative activity of cancer cells in vitro. For this, we used the HCC1806, MDA-MB-231 (triple-negative breast cancer [TNBC]), H1299 (non-small cell lung cancer [NSCLC]), and PC-3 prostate cancer cell lines. We observed that 2-ANPC significantly affected growth kinetics in the aforementioned cancer cell lines (Figure 1). Notably, this effect was dose-dependent and more potent than that of paclitaxel (PTX), a potent MTA agent used as a positive control.

Besides the anti-proliferative effect of 2-ANPC in epithelial cancer cells, this amino-pyrrole derivative exhibited potent pro-apoptotic activity, which was evidenced by the increased expression of common apoptotic markers (e.g., cleaved forms of poly(ADP)-ribose polymerase (PARP) and caspase-3) at 48 h post-treatment (Figure 2).

### 3.2. 2-ANPC Effectively Decreases HIF-1α Expression In Vitro in Epithelial Cancer Cells by Promoting Its Proteasome-Dependent Degradation

Besides the anti-proliferative and pro-apoptotic activities of 2-ANPC in the aforementioned cancer cell lines, this 2-aminopyrrole derivative effectively decreased the expression of HIF-1α in the majority of cancer cell lines used in the present study. This was shown for both HCC1806 and MDA-MB-231 TNBC cells and PC-3 cells, whereas H1299 NSCLC cells exhibited a moderate decrease in HIF-1α after 2-ANPC treatment (Figure 3A). In concordance with WB data, 2-ANPC also effectively decreased HIF-1α at the transcriptional level. Indeed, real-time polymerase chain reaction (RT-PCR) data shown in Figure 3B illustrate a significant decrease in HIF-1α mRNA levels after 2-ANPC treatment of HCC1806, MDA-MB-231, and PC-3 cells. Again, no significant difference in HIF-1α mRNA levels between non-treated and 2-ANPC-treated cells was observed in H1299 lung cancer cells (Figure 3B).

Next, we examined whether 2-ANPC affects HIF-1α’s stability. To test this possibility directly, we treated PC-3 prostate cancer cells with 2-ANPC in the presence of MG-132, a 26S proteasome inhibitor. The data shown in Figure 3C illustrate increased expression of HIF-1α in cancer cells cultured with 2-ANPC in the presence of MG-132, compared with cancer cells cultured with 2-ANPC alone, thereby suggesting rapid protein turnover in cancer cells after exposure to 2-ANPC due to increased proteasome-mediated HIF-1α degradation. Overall, this data shows that this 2-aminopyrrole derivative significantly alters HIF-1α stability at the protein level and promotes its proteasome-dependent degradation.

To examine whether 2-ANPC exhibits an inhibitory effect on HIF-1α in conditions relevant to in vivo conditions, we performed a cell-based HIF-1α stabilization assay under hypoxic conditions. For this purpose, we performed experiments in a cobalt chloride (CoCl2)-modeled hypoxic environment and observed upregulation of HIF-1α at the protein level in almost all cancer cell lines used in the present study. Indeed, we found that 2-ANPC induced a moderate decrease in HIF-1α expression under hypoxic conditions, thereby revealing its impact on HIF-1α signaling in cancer cells (Figure 4A,B). We also assessed HIF-1α at the transcriptional level and observed a similar mRNA HIF-1α pattern of hypoxic induction with and without 2-ANPC (Figure 4C). Overall, this data illustrates the high potency of 2-ANPC in targeting HIF-1α and regulating its signaling in cancer cells under hypoxic conditions.

### 3.3. 2-ANPC Inhibits Tumor Growth and Decreases HIF-1α, VEGFR1, and VEGFR3 Expression In Vivo

Next, we examined 2-ANPC for its antitumor and HIF-1α-inhibitory activities using the 4T1 breast cancer syngraft model. Syngraft tumors were allowed to reach ~200 mm^3^ before randomization of mice into two groups (control (*n* = 5) and 2-ANPC-treated (*n* = 5)) (day 10 after inoculation). 2-ANPC was diluted as described and administered on days 10, 13, 16, 19, 22, 25, and 28 after inoculation. The dose of 2-ANPC used for this experiment was 10 mg/kg. We observed a moderate decrease in tumor volume and weight in 2-ANPC-treated mice bearing 4T1 syngrafts when compared with vehicle-treated (control) mice, illustrating that 2-ANPC exhibits antitumor activity. Importantly, no toxicity was observed in any animal group throughout the experiment (30 days). The graphs depicting the decrease in tumor volume and weight after 2-ANPC treatment are shown in Figure 5A and Figure 5B, respectively. To examine whether the increased apoptosis in 4T1 syngrafts was responsible for the aforementioned changes in tumor volumes and weights, we performed immunohistochemical (IHC) staining of tumors for cleaved caspase-3. Strikingly, we observed a significant increase in the number of caspase-3-positive (i.e., apoptotic) cells after 2-ANPC treatment, as shown in Figure 5C.

In concordance with these findings, 2-ANPC-treated syngrafts also exhibited an increase in areas of central necrosis as assessed by hematoxylin and eosin (H&E) staining when compared with control (solvent-treated) 4T1 syngrafts (Figure 6A). Importantly, the increased number of pathological mitoses was also detected in 2-ANPC-treated syngrafts (Figure 6B), thereby revealing the previously shown molecular mechanism of action of this derivative, targeting the microtubule network due to tubulin depolymerization and inducing substantial changes in cell cycle regulation and accumulation of cancer cells in the M-phase [18,19].

Based on our in vitro data illustrating 2-ANPC’s potency in decreasing HIF-1α in cancer lines (as shown in Figure 3), we examined whether the antitumor activity of this amino-pyrrole derivative was due to its ability to decrease HIF-1α expression in vivo. Strikingly, we observed reduced HIF-1α expression in 4T1 syngrafts after 2-ANPC treatment (Figure 7A,B). This data was also consistent with in vitro data demonstrating a moderate decrease in HIF-1α expression in 4T1 cells cultured in the presence of 2-ANPC (Figure 7C). Again, this derivative effectively inhibited the proliferative activity of this cancer cell line and was much more effective than PTX, used as a positive control for these experimental settings (Appendix A).

In concordance with the 2-ANPC-induced decrease in HIF-1α in 4T1 syngrafts, the expression of VEGFR1 and 3 was also reduced after 2-ANPC treatment when compared with vehicle-treated controls (Figure 8A–D), thereby revealing hypoxia in the tissues as a potent mechanism promoting angiogenesis.

### 3.4. Molecular Modeling Studies

#### 3.4.1. Folding and Binding Site Search

First, we considered folding HIF-1α alone; its structure was predicted using ColabFold. As expected, the best folded domains were basic helix–loop–helix (bHLH) and Per-Arnt-Sim (PAS) (Figure 9), which were partially included in the protein data bank (PDB) 4H6J and PDB 4ZPR crystals.

The unfolded regions were identified as the ODD domain (401–603), the Inhibitory Domain (ID, 576–785), and the c-terminal transactivation domain (CTAD) (786–826), which could be assigned to intrinsically disordered regions (IDRs). Thus, the authors of [42] proposed that the ODD and Inhibitory Domain (ID) were intrinsically disordered, in agreement with the Nuclear Magnetic Resonance (NMR) and far-UV circular dichroism data. NMR demonstrated the disorder character of CTAD, as shown in [43].

The HIF-1α model was equilibrated for 100 ns in a molecular dynamics simulation (Appendix A). The system snapshot at 99.6 ns was selected from the stable-trajectory cluster based on the fine Ramachandran plot (Appendix A). After the protein minimization, the system was used for the subsequent investigation.

#### 3.4.2. Site Mapping and Multi-Ligand Dynamics

By searching for potential binding sites using SiteMap, five potential cavities were found (Figure 10). Thus, two possible sites were identified in the bHLH and ID domains and two in the PAS and ODD domains, with the latter specific to the ODD domain. Unfortunately, the following docking did not confirm the latter of the found sites as potentially binding, so we considered that we had only seen the four sites at this step. For additional site searching, we performed multi-ligand molecular dynamics, in which 26 copies of the most active 2-ANPC were placed in the simulation box. After 300 ns, the trajectories of each small molecule were analyzed, and potential sites were identified. It should be noted that not all of the protein surface interacted with ligands; generally, the PAS domain did not bind to 2-ANPC. That approach enabled us to identify additional binding sites within the structural domain of HIF-1α, outside any of its functional domains. The ligand occupied this site for almost the entire simulation, with only minor conformational changes.

The 2-ANPC was docked into each site. As mentioned above, the one site identified by SiteMap did not yield any plausible docking poses, so we did not consider it further. For sites 1–5, the docking scores from −5.9 to −9.8 kcal/mol were obtained (Appendix A).

Encouraging data were obtained for sites 1–4. Based on the protein structure, binding at two positions could directly affect HIF-1α activity by interacting with the bHLH domain. By binding to sites 1 and 4, 2-ANPC could prevent dimerization of HIF-1α with HIF-1β as it lies near the PAS-A and B domains. Interestingly, one of the good binding sites, 3, was partially formed by the ODD domain, which could potentially inhibit HIF-1α degradation, leading to a reversal of the effect and HIF-1α accumulation. At the same time, while the ODD domain is involved in active HIF complex formation, where its secondary and tertiary structure would be stabilized, its stabilization by the ligand may have a counterintuitive effect. The fifth site, identified from multi-ligand dynamics, was located partially within the PAS domain and partially in the disordered region. Binding the ligands could also prevent active complex formation, but we obtained a poor docking score for that site.

Considering the protein pocket structures, it is notable that sites 1 and 4 were well-folded, whereas sites 3 and 2 contained significant unfolded regions. This could indicate that sites 1 and 4 were the result of false-positive site mapping and docking. On the other hand, the binding and stabilization of the unfolded polyprotein could prevent dimerization with HIF-1β and the formation of an active complex, suggesting that its ligands act as protein–protein interactions (PPIs). Nevertheless, in vitro experiments and more complex modeling should be conducted to confirm or refute the existence of these binding sites.

Interestingly, the second site contains Ser 641. If the ligand actually stabilizes the ODD region close enough to Ser641, it could prevent its phosphorylation by mitogen-activated protein kinase, leading to more efficient chromosome region maintenance 1 (CRM1)-dependent nuclear export of the CRM1 and less potent HIF-1 activity [44]. On the other hand, pocket 2 contains a part of the bHLH domain, in particular Lys32, for which a significant role in regulating HIF-1α activity was demonstrated. It could be proposed that inhibiting the DNA-binding domain would decrease HIF-1 activity. Nevertheless, it was described that the methylation of Lys32 serves as a signal for the recruitment of the ubiquitin ligase complex that targets HIF-1α for proteasomal degradation via a VHL-independent pathway [45]. As a result, inhibition of Lys34 methylation could enhance HIF-1 activity.

Moreover, due to the complex role of HIF-1α and its regulation of activity, it is difficult to say whether the other domains or essential regulatory amino acids are influenced by binding to the found sites. We considered only the ligand-binding regions of these pockets and proposed the corresponding effects on HIF-1 activity. Nevertheless, the impact on HIF-1α regulatory activity or protein interactions could be indirect, i.e., through allosteric regulation or the PPI pathway, and it is not possible to predict the result without direct cellular experiments.

In summary, we performed a folding analysis of HIF-1α alone and identified six potential binding sites, four of which yielded favorable docking scores. Two of these sites were located in a well-folded region.

### 3.5. HIF Active Complex Binding Hypothesis

Considering the second hypothesis, which assumes the compound acts as a DNA-binding inhibitor, the entire HIF-1 protein complex should be taken into account. It follows from the complex mechanism of action as well as the HIF-1α disorder structure; thus, if the ligands bind site 2 of HIF-1α, which is mostly formed by the unfolded polypeptide chain, it would possibly prevent dimerization with HIF-1β rather than direct HIF-1 activity.

#### 3.5.1. Folding and Molecular Dynamics

The most complete modeling of the HIF-1α-HIF-1β complex was performed in [46]. The authors did not describe the obtained HIF-1α-HIF-1β complex in detail, but, based on [44], it can be concluded that the PAS and bHLH domains of the proteins are separated, which contradicts the experimental data from the protein crystals 4ZPR and 4H6J.

In an attempt to fold the HIF-1 protein, it was found that HIF-1α and HIF-1β, when folded together, resulted in both well-folded and unfolded regions, similar to what was observed with HIF-1α alone. Surprisingly, simultaneous folding of HIF-1α, HIF-1β, and p300 together led to a fine folded structure, where disordered regions were found to be packed into alpha-helices. Moreover, the DNA fragment to be added to the complex for folding was placed directly into the bHLH domain formed by HIF-1α and HIF-1β.

It should be noted that system growth, which initially contains disordered sequences, can lead to false positives due to a significant increase in local energy minima [46]. For this reason, we chose the complex with mutual spatial arrangement of HIF-1α and p300 chains, as obtained in 7LVS [47] for HIF-1α and CPB. An attempt to include Zn^2+^ in the model during the folding stage did not yield the correct ion positions according to 7LVS, so we used a complex without ions for further complex stabilization via a 300 ns molecular dynamics simulation (Appendix A). As additional validation, it was considered that both PAS domains of HIF-1α and HIF-1β were well-folded, and their interface was consistent with the PDB 4H6J crystal structure. Moreover, the folded conformations of HIF-1α and HIF-1β parts, which formed the bHLH domain, were consistent with the corresponding structure of the 4ZPR PDB crystal. Based on this, we considered the obtained protein complex model to be more accurate (Appendix A) than the model, as shown in [48].

Interestingly, IDRs in protein complexes are often organized as alpha-helices, making it difficult to assess the reliability of the folded model. Nevertheless, the well-folded regions of these three proteins were consistent with the corresponding crystal structures. After the folding and 300 ns relaxation dynamics, we selected the most suitable snapshot based on trajectory clustering and analysis of the Ramachandran plots (Appendix A).

#### 3.5.2. Site Mapping and Docking of HIF-1α and HIF-1β

After the folding, the site mapping was performed. We proposed only the DNA-binding inhibitory mechanism, excluding any far-allosteric sites, due to the system size and model complexity, as well as the possibility that the folded complex could contain numerous inaccuracies that could not be resolved without experimental data.

Thus, bHLH domains of HIF-1α and HIF-1β recognize and bind DNA, while the bHLH of HIF-1β is not the terminal protein domain. As demonstrated by folding, the downstream segment is located on the complex surface, suggesting its conformational flexibility and potential influence on DNA binding. Accordingly, we considered not only the bHLH domain of HIF-1α but also those of both HIF-1α and HIF-1β. Binding these domains could prevent ligands from directly binding DNA, or at least stabilize both a-helices, preventing them from assuming the conformation required for DNA interaction. Unfortunately, it was not specified whether the triple protein complex formed alone before DNA binding, or whether HIF-1α and HIF-1β first bind to the DNA, with the activator subsequently binding. Therefore, we attempted to model the first mechanism of inhibition. Using the SiteMap with the following docking, we identified two sites that contain bHLH domains in both proteins. The other four sites were found to be located closer to the N-terminus of HIF-1β (Figure 11). Due to the uncertainty of the HIF-1β conformations in these pockets, despite the good docking scores of 2-ANPC, we considered sites 2–6 as more questionable.

Considering the docking scores (Appendix A), as well as the folding results and partial validation by the PDB crystal structure, we concluded that only one site is more reliable for ligand binding within the whole active complex. Despite this, the two sites include both the HIF-1α and HIF-1β fragments and plausible docking results; however, we considered them less favorable due to their location in a significantly flexible and poorly pre-organized region.

## 4. Discussion

It is well known that HIF-1α is a common regulator of a broad spectrum of molecular pathways in solid tumors. In particular, this includes angiogenesis, glycolysis, cancer cell proliferation, growth, migration, and metastasis formation [49,50,51,52].

Besides the activities above, hypoxia can promote the progression of human malignancies by forming an immunosuppressive TME via the activation of immunosuppressive cells and the inhibition of the activities of cytotoxic T cells, including CD8 T cells and NK cells [53,54,55].

Additionally, the HIF-1α signaling pathway plays a vital role in metabolic reprogramming in solid tumors, including enhancing glucose uptake by positively regulating glycolytic enzymes. This pathway also regulates the expression of pyruvate dehydrogenase kinase, a well-known negative regulator of pyruvate dehydrogenase, leading to the conversion of pyruvate to lactate rather than acetyl-CoA. Notably, HIF-1α increases reactive oxygen species levels in cancer cells by inhibiting the tricarboxylic acid cycle and activating the pentose phosphate pathway [52,56].

Lastly, HIF-1α might also be involved in cancer resistance to chemo- and radiotherapies. Indeed, under hypoxic conditions, solid tumors rely on glycolysis for energy production, which results in an acidic microenvironment, making them less sensitive to chemotherapy and radiotherapy [57].

Therefore, targeting HIF-1α and its related proteins is currently considered a promising approach for developing new anticancer therapeutics and sensitizing chemo- and radioresistant malignancies to current therapies [4,55,58,59].

We show here for the first time that 2-aminopyrrole derivative 2-ANPC exhibits high potency in interacting with HIF-1α and downregulates its expression in multiple cancer cell lines in vitro. Our findings were further confirmed by in vivo studies showing a significant decrease in HIF-1α expression in breast cancer syngrafts. Notably, this decrease was also associated with reduced expression of VEGFR1 and 3 and correlated with the decreased tumor volumes after 2-ANPC treatment. Moreover, we found that 2-ANPC effectively downregulated HIF-1α at both the translational and transcriptional levels in the majority of cancer cells and promoted its proteasome-dependent degradation.

Among known small molecules that interact with HIF-1α and are relevantly similar to the studied 2-aminopyrrole derivatives are 2-methoxyestradiol, lificiguat, and acriflavine.

2-Methoxyestradiol (2-ME2) is a metabolite of estradiol possessing HIF-inhibiting activity [60]. Inhibition of hypoxia-inducible factor by 2-ME2 eliminates the previously described effects of HIF-1α expression, thereby reducing the growth rate of the primary tumor and, consequently, its angiogenesis. Equally important is the overcoming of tumor resistance to chemotherapy [60,61].

Lificiguat (YC-1) is a small molecule whose heterocyclic structure is known as one of the factors contributing to its HIF-inhibiting activity [62,63]. Acriflavine—an acridine dye with antiseptic properties—has demonstrated HIF-1-inhibiting properties, manifested in reduced tumor angiogenesis and an improved response to radiotherapy in brain tumors [64]. The aforementioned HIF inhibitors exhibit structural similarities to the 2-aminopyrrole derivative, 2-ANPC, used in the present study. Given the small size of the molecules, the presence of heterocyclic nitrogen-containing molecules, such as acriflavine and YC-1, appears significant. Furthermore, the biological activities of 2-aminopyrroles correlate with the action of 2-ME2. Indeed, 2-ANPC exhibited potent cytotoxic activities against a broad spectrum of epithelial cancer cell lines, including breast, lung, and prostate cancer. The anticancer activity of 2-ANPC was due to its ability to disrupt the microtubule network and inhibit tubulin polymerization [18,19]. In addition to strong cytotoxic and anti-proliferative activities of 2-ANPC, this 2-aminopyrrole derivative exhibited potent anti-tumor activity in a xenograft tumor model (HCC1806 cell line), demonstrating results comparable to those of paclitaxel used as a positive control for in this study.

Molecular docking demonstrated efficient binding of 2-ANPC to the colchicine-binding site on tubulin [18,19]. Similar interaction patterns and biological activities were observed for 2-ME2 [65,66]. Besides binding to tubulin, 2-ME2 binds to the PAS-B domain of HIF-1α [67]. Thus, the examination of the HIF-1-inhibitory activities of 2-ANPC, with proven cytotoxic, anti-proliferative, and anti-tumor activities, is highly attractive and, if successful, would lead to 2-aminopyrroles being considered a novel class of potent anticancer agents with a dual mode of action, effectively targeting microtubule polymerization and inhibiting HIF-1 activity.

HIF-1α, HIF-1β, and CBP/p300 are nuclear proteins with multiple functions and contain intrinsically disordered regions. Only a few crystal structures of HIF-1α fragments have been reported; for example, PDB ID 4H6J [68] contains the PAS-B domain of the HIF-1α/HIF-1β heterodimer, and PDB ID 4ZPR [69] includes parts of the PAS-A, PAS-B, and bHLH domains from both subunits. Several crystal structures contain short sequences of HIF-1α, i.e., complexes of Factor Inhibiting HIF-1α or pVHL that selectively hydroxylate amino acids (Asn803 by HIF-1 or Pro402/Pro564 by pVHL), thereby inhibiting the activity of HIF-1α. Protein crystals up to 51 amino acids of HIF-1α are thereby presented. In this line of evidence, and given our current data showing 2-ANPC-induced decreased expression of HIF-1α (as shown in Figure 3 and Figure 4), further studies using VHL knockdown with corresponding siRNA or VHL inhibitors (e.g., VH032 and VH298) will be very informative to determining whether 2-ANPC still reduces HIF-1α under VHL inhibition conditions.

Thus, considering HIF-1α as a molecular target, based on its domain functions and its mechanism of action, we identified two possible molecular mechanisms and consequences of its interactions with 2-ANPC. First, small molecules could bind to domains that mediate protein–protein interactions, thereby preventing complex assembly. In particular, binding compounds to the PAS or CTAD of proteins could act as PPI inhibitors. Second, binding the bHLH domain of the entire complex prevents ligands from binding DNA and inhibits HIF-1 activity. This might also interfere with HIF-1α’s stability and promote its degradation, thereby leading to a significant decrease in HIF-1α expression in 2-ANPC-treated cancer cells. Moreover, significant downregulation of HIF-1α expression in 4T1 syngrafts in 2-ANPC-treated mice correlated with a substantial reduction in the expression of VEGFR1 and 3, thereby illustrating that inhibition of HIF-1α signaling as a secondary anti-tumor mechanism of 2-ANPC, supplementing its potent cytotoxic and anti-proliferative activities in vivo.

Overall, we show here for the first time that 2-ANPC, the 2-aminopyrrole derivative synthesized in our lab and previously shown to be a potent microtubule-targeting compound that induces tubulin depolymerization [18,19] also exhibits potent anti-HIF-1α activity. Besides the direct targeting of HIF-1α, a significant decrease in HIF-1α at both transcriptional and translational levels in 2-ANPC-treated cancer cells might be due to its previously demonstrated ability to alter microtubule dynamics and inhibit tubulin polymerization. Indeed, several reports indicate that translational initiation of HIF-1α mRNA is also regulated by microtubule dynamics, and the disruption of microtubule dynamics suppresses HIF-1α mRNA translation and leads to the accumulation of HIF-1α mRNA in the P-bodies [70]. For example, 2-ME, a derivative of estradiol that lacks estrogenic activity, was shown to inhibit microtubule polymerization and translation of HIF-1α mRNA. This was associated with potent anti-tumor activity in vitro and in vivo, as demonstrated using various xenograft tumor models [68,71].

Collectively, our data shown here illustrate that 2-ANPC can serve as a scaffold for the development of successful chemotherapeutic anticancer agents with dual therapeutic modalities.

## 5. Conclusions

Overall, we show here that 2-ANPC, the 2-aminopyrrole derivative synthesized in our lab and previously shown to be a potent microtubule-targeting compound that induces tubulin depolymerization, also exhibits potent anti-HIF-1α activity. Besides the direct targeting of HIF-1α, a significant decrease in HIF-1α at both transcriptional and translational levels in 2-ANPC-treated cancer cells might be due to its previously demonstrated ability to alter microtubule dynamics and inhibit tubulin polymerization. This in turn illustrates that 2-ANPC can serve as a scaffold for the development of successful chemotherapeutic anticancer agents with dual therapeutic modalities.

## Data Availability

The original contributions presented in the study are included in the article/Appendix A; further inquiries can be directed to the corresponding author.

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
