# Peer review of "Hypoxia-Inducible Factor-1α, a Novel Molecular Target for a 2-Aminopyrrole Derivative: Biological and Molecular Modeling Study"

_cancers, 2025, doi:10.3390/cancers18010115_

Round 1
Reviewer 1 Report
Comments and Suggestions for Authors
The manuscript investigates the anticancer effects of the 2-aminopyrrole compound 2-ANPC, proposing that it acts as both a microtubule inhibitor and a suppressor of HIF-1α. The topic is of clear interest and relevance, and the authors present a broad range of experiments, including molecular modeling, in vitro assays, and an in vivo syngraft model.
While the study presents promising preliminary findings, several of the central claims—particularly regarding direct targeting of HIF-1α and the mechanistic basis of the observed effects—require additional experimental validation. In its current form, the manuscript is not yet suitable for publication, but it has the potential to become a strong contribution if the issues below are addressed.
The conclusion that 2-ANPC interacts with HIF-1α is primarily supported by Western blot showing reduced HIF-1α, mRNA decrease, proteasome inhibitor (MG132) effects, in silico docking & MD simulations. However, lack of data under hypoxic condition and HIF-1α activity weaken the significance of this study. Therefore, I recommend adding the data obtained by cell-based HIF-1α stabilization assay using cobalt chloride or DMOG, and HIF-1α transcriptional activity assay.
The authors suggest proteasomal degradation, but the pathway is not shown. So it’s better to investigate the effect of 2-ANPC in the condition of VHL knockdown (siRNA) or VHL inhibitor study. If 2-ANPC still reduces HIF-1α under VHL inhibition, alternative pathways could be considered.
Author Response
We thank very much the reviewer for the detailed analysis of our manuscript and comments and suggestions, as well. Below are our specific responses to reviewer’s comments (shown in quotes and italics). The changes in the revised version of the manuscript are highlighted with yellow.
Comments 1: “The manuscript investigates the anticancer effects of the 2-aminopyrrole compound 2-ANPC, proposing that it acts as both a microtubule inhibitor and a suppressor of HIF-1α. The topic is of clear interest and relevance, and the authors present a broad range of experiments, including molecular modeling, in vitro assays, and an in vivo syngraft model.”
Response 1: We appreciate reviewer for a detailed analysis of our work and a high assessment of the scientific results presented in the article.
Comments 2: “While the study presents promising preliminary findings, several of the central claims—particularly regarding direct targeting of HIF-1α and the mechanistic basis of the observed effects—require additional experimental validation. In its current form, the manuscript is not yet suitable for publication, but it has the potential to become a strong contribution if the issues below are addressed. The conclusion that 2-ANPC interacts with HIF-1α is primarily supported by Western blot showing reduced HIF-1α, mRNA decrease, proteasome inhibitor (MG132) effects, in silico docking & MD simulations. However, lack of data under hypoxic condition and HIF-1α activity weaken the significance of this study. Therefore, I recommend adding the data obtained by cell-based HIF-1α stabilization assay using cobalt chloride or DMOG, and HIF-1α transcriptional activity assay”.
Response 2: We greatly appreciate the reviewer for the detailed analysis of the data presented in our manuscript and for the comments and suggestions. We totally agree with the reviewer that the experimental data from a cell-based HIF-1α stabilization assay under hypoxic conditions will strengthen our manuscript and provide direct evidence of the impact of 2-ANPC on these mechanisms. For this purpose, we performed experiments in a cobalt chloride (CoCl2)-modelled hypoxic environment and observed upregulation of HIF-1α at the protein level in all cancer cell types used in our study, as shown in Figure 4. Important, 2-ANPC induced a moderate decrease in HIF-1α expression under hypoxic conditions, thereby revealing its impact on HIF-1α expression and its signaling in cancer cells. This western blot data was included in the revised manuscript and is currently shown in Figure 4A. Densitometric analysis of this data was also included in the revised manuscript and shows significant differences in HIF-1α expression between cancer cells treated with cobalt chloride in the absence or presence of 2-ANPC (Figure 4B). We also assessed changes in HIF-1α expression at the transcriptional level under similar experimental conditions. As shown in Figure 4C, a similar pattern of HIF-1α with and without 2-ANPC was observed in both types of triple-negative breast cancer cells (e.g., HCC1806 and MDA-MB-231). Overall, in our opinion, this data illustrates the high potency of 2-ANPC in targeting HIF-1α and regulating its signaling in cancer cells.
Comments 3: “The authors suggest proteasomal degradation, but the pathway is not shown. So it’s better to investigate the effect of 2-ANPC in the condition of VHL knockdown (siRNA) or VHL inhibitor study. If 2-ANPC still reduces HIF-1α under VHL inhibition, alternative pathways could be considered”
Response 3. We are greatly thankful to a reviewer for this suggestion. Unfortunately, we do not have the opportunity to perform a VHL knockdown (siRNA) or use a VHL inhibitor due to the limited time frame for the revision (10 days) and technical limitations related to the absence of the corresponding siRNA and VHL inhibitors. We described the importance and scientific value of these experiments in the Discussion. To strengthen our data on 2-ANPC-induced proteasomal degradation of HIF-1a, we show here the increased expression of HIF-1a in MG-132-treated cells cultured in presence of 2-ANPC, thereby suggesting proteasomal degradation of HIF-1α induced by the 2-aminopyrrole derivative.
After addressing these issues, we are resubmitting our revised manuscript for your kind consideration.
Sincerely,
Sergei Boichuk
Reviewer 2 Report
Comments and Suggestions for Authors
Dear Authors,
This is a well-designed and comprehensive study demonstrating that the 2-aminopyrrole derivative 2-ANPC represents a novel molecular modulator of HIF-1α. The authors combine in vitro analyses across multiple epithelial cancer cell lines, in vivo validation using a 4T1 syngeneic model, and detailed molecular modeling approaches, which together provide a coherent and convincing mechanistic framework. The experimental data are generally robust, well controlled, and clearly support the main conclusions of the study.
Only minor revisions are suggested to further improve clarity and presentation:
- Figure legends and quantification:
For Western blot analyses (e.g., Figures 2, 3, and 6), the authors may consider clarifying in the figure legends whether the densitometric values represent averages from independent experiments and indicating the corresponding sample size (n), where applicable. - HIF-1α mRNA vs protein regulation:
In H1299 cells, the differential effect of 2-ANPC on HIF-1α protein versus mRNA levels is interesting. A brief discussion of possible post-transcriptional or post-translational regulatory mechanisms in the Discussion section would strengthen the interpretation. - In vivo experimental details:
While the in vivo methodology is adequately described, consolidating the group sizes and statistical information in the Materials and Methods section could improve readability. - Molecular modeling interpretation:
The molecular docking and dynamics analyses are extensive and informative. Some statements describing ligand–protein interactions could be phrased slightly more cautiously (e.g., “putative” or “potential” binding sites) to reflect the predictive nature of the modeling. - Language and style:
The manuscript is generally well written; however, minor editorial refinements could further improve clarity in a few long or complex sentences.
Overall, these minor points do not affect the scientific validity of the work. After addressing the comments above, the manuscript will be suitable for publication.
Best regards,
Reviewer
Comments on the Quality of English LanguageThe manuscript is generally well written and clearly understandable. Minor editorial improvements could be made to enhance clarity and flow in some sections, particularly by simplifying a few long sentences. These are stylistic issues and do not affect the scientific content or interpretation of the results.
Author Response
We thank very much the reviewer for the detailed analysis of our manuscript and comments and suggestions, as well. Below are our specific responses to reviewer’s comments (shown in quotes and italics). The changes in the revised version of the manuscript are highlighted with yellow.
Comment 1: "
“This is a well-designed and comprehensive study demonstrating that the 2-aminopyrrole derivative 2-ANPC represents a novel molecular modulator of HIF-1α. The authors combine in vitro analyses across multiple epithelial cancer cell lines, in vivo validation using a 4T1 syngeneic model, and detailed molecular modeling approaches, which together provide a coherent and convincing mechanistic framework. The experimental data are generally robust, well controlled, and clearly support the main conclusions of the study. Only minor revisions are suggested to further improve clarity and presentation. The manuscript is generally well written and clearly understandable. Minor editorial improvements could be made to enhance clarity and flow in some sections, particularly by simplifying a few long sentences. These are stylistic issues and do not affect the scientific content or interpretation of the results.”
Response 1: We greatly appreciate reviewer for a detailed analysis of our work and a high assessment of the scientific results presented in the article.
Comment 2: "“Figure legends and quantification:
For Western blot analyses (e.g., Figures 2, 3, and 6), the authors may consider clarifying in the figure legends whether the densitometric values represent averages from independent experiments and indicating the corresponding sample size (n), where applicable”
Response 2: We appreciate a reviewer for this notice. All the WB experiments were performed in triplicate, and the corresponding sample size (n=3) was included in the Figure legends in the revised manuscript.
Comment 3: "HIF-1α mRNA vs protein regulation: In H1299 cells, the differential effect of 2-ANPC on HIF-1α protein versus mRNA levels is interesting. A brief discussion of possible post-transcriptional or post-translational regulatory mechanisms in the Discussion section would strengthen the interpretation”.
Response 3: We greatly appreciate a reviewer for this notice and suggestion. Indeed, H1299 NSCLC cells were much less sensitive to the 2-ANPC-induced inhibition of HIF-1α expression when compared with TNBC cell lines and PC-3 cells, as well. Of note, this was observed at both the protein and mRNA levels, as shown in Figures 3A and 3 B, respectively. We observed a minor (less than 20%) reduction in HIF-1α protein expression after 2-ANPC treatment, whereas HIF-1α mRNA levels were not changed in these experimental settings. This might be due to decreased stability of the HIF-1α protein in H1299 cells and increased proteasome degradation. These possibilities were addressed in the Discussion of the revised manuscript. In all remaining cases, we observed a strong correlation pattern between decreased levels of HIF-1α at both the protein and mRNA levels.
Comment 4: “In vivo experimental details: While the in vivo methodology is adequately described, consolidating the group sizes and statistical information in the Materials and Methods section could improve readability”.
Response 4: We are thankful to the reviewer for this suggestion and have included information on group sizes and statistical methods in the Materials and Methods section.
Comment 5: “Molecular modeling interpretation: The molecular docking and dynamics analyses are extensive and informative. Some statements describing ligand–protein interactions could be phrased slightly more cautiously (e.g., “putative” or “potential” binding sites) to reflect the predictive nature of the modeling”.
Response 5: We appreciate the reviewer's notice and agree with it. We rephrased several statements during the interpretation of molecular modelling data. These changes are also highlighted in yellow in the revised manuscript.
Comment 6: “Language and style: The manuscript is generally well written; however, minor editorial refinements could further improve clarity in a few long or complex sentences”.
Response 6: We also agree with this suggestion and appreciate it. Thus, we rephrased some long or complex sentences to make them clearer and easier to understand for readers.
After addressing these issues, we are resubmitting our revised manuscript for your kind consideration.
Sincerely,
Sergei Boichuk
Round 2
Reviewer 1 Report
Comments and Suggestions for Authors
The author responded to my suggestions satisfactory and now the revised manuscript sounds more impressive. Therefore, I recommend it for publication.